# Community Forums as Amplifiers of Communities’ Voices: Isolated Communities in Puerto Rico

**DOI:** 10.3390/ijerph20146335

**Published:** 2023-07-10

**Authors:** Yashira M. Sánchez Colón, Edna Acosta Pérez, Mayra L. Roubert Rivera, Marizaida Sánchez Cesáreo, Christine Miranda Diaz, Glenda L. Ortiz, Jean C. Meléndez González, Valeria M. Schleier Albino, Laura Mora Lemus

**Affiliations:** 1Public Health Program, Ponce Health Sciences University, Ponce Research Institute-Ponce Medical School Foundation, Inc., Hispanic Alliance for Clinical and Translational Research, Ponce 00716, Puerto Rico; mroubert@psm.edu; 2Center for Sociomedical Research and Evaluation, Graduate School of Public Health, University of Puerto Rico Medical Sciences Campus, Hispanic Alliance for Clinical and Translational Research, San Juan 00936, Puerto Rico; edna.acosta2@upr.edu (E.A.P.); marizaida.sanchez@upr.edu (M.S.C.); 3Grupo Nexos, Inc., San Juan 00936, Puerto Rico; jcmelendez@nexospr.org; 4Institute of Research, Education, and Services in Addiction (IRESA), Internal Medicine Department, Universidad Central del Caribe, Hispanic Alliance for Clinical and Translational Research, Bayamón 00956, Puerto Rico; christine.miranda@uccaribe.edu; 5Clinical Research Center, Universidad Central del Caribe, Hispanic Alliance for Clinical and Translational Research, Bayamón 00956, Puerto Rico; 6Environmental Health Department, Graduate School of Public Health, University of Puerto Rico Medical Sciences Campus, Hispanic Alliance for Clinical and Translational Research, San Juan 00936, Puerto Rico; 7Community Health and Research Council for the Hispanic Alliance for Clinical and Translational Research, San Juan 00936, Puerto Rico; laura.mora567@gmail.com

**Keywords:** isolation, community forum, social determinants of health, health priorities, Puerto Rico, Hispanic Alliance for Clinical and Translational Research (Alliance)

## Abstract

Social determinants of health contribute to health disparities and inequities. We conducted a community forum on the topic of isolation with the objectives of (1) identifying and prioritizing key health-related issues needing attention in isolated communities in Puerto Rico; (2) developing strategies in terms of Policies, Programs, and Practices to address the community priorities we identified. We used the triangulation method for qualitative data, integrating the Colorado State University’s Tri-ethnic Center Model and the Delphi Technique for a better understanding of community health needs and priorities. The five community health-related priorities identified in the community forum were: (1) access to health services (physical and mental); (2) older adults; (3) access to basic services; (4) preparedness for future disasters/emergencies; and (5) COVID-19 and access to vaccination. The Alliance Leaders and Advisory Boards understand that we will work with the priorities of preparedness for future natural disasters/emergencies and COVID-19 and access to vaccination. Fifteen strategies were developed for these priorities and were grouped into five areas that require more attention in order to reduce health disparities. Isolated communities in Puerto Rico present an intersectionality of factors that affect a wide range of health-related risks and outcomes.

## 1. Introduction

Social determinants of health (SDOH) are the social conditions in which people are born, grow, live, learn, work, play, and age that contribute to health disparities and inequities [1]. Healthy People 2030 organized the SDOH into five categories: (1) economic stability; (2) education access and quality; (3) social and community context; (4) healthcare access and quality; and (5) neighborhood and built environment [1]. The Healthy People initiative has been released by the United States (US) Department of Health and Human Services every decade since 1980 to guide national health promotion and disease prevention efforts to improve the health of the nation [2].

For certain health conditions, Hispanics bear a disproportionate burden of disease, injury, death, and disability due to social determinants compared to non-Hispanic whites, the largest racial/ethnic population in the US [3]. Hispanics or Latinos are the largest racial/ethnic minority group in the US [4] and refer to a person of Cuban, Mexican, Puerto Rican, South or Central American, or other Spanish culture or origin regardless of race [5].

Disparities in income, education, occupational status, or wealth are translated into a measure of social hierarchy called socioeconomic status (SES) [6,7]. SES has been viewed as the driver of discrimination and racial inequities in health [7]. Healthcare discrimination occurs at both individual and structural levels, due to race, gender, and other elements [7,8]. Discrimination at the individual level includes negative interactions between a patient and a healthcare provider and structural discrimination is seen in the form of residential segregation for geographic and economic reasons [7,8]. According to an intersectionality perspective, health inequities are the results of different social locations (e.g., race/ethnicity, Indigeneity, gender, class, sexuality, geography, age, disability/ability, migration status, religion), power relations (e.g., laws, policies, state governments and other political and economic unions, religious institutions, media), and experiences [9]. Additionally, historical trauma suggests that populations historically subjected to long-term, collective trauma (or traumatic experience) transmit these traumas to subsequent generations, causing them a higher prevalence of adverse effects, including an impact on physical and mental health [10].

Currently, Puerto Rico is experiencing health disparities in available health services and health outcomes. The Commonwealth of Puerto Rico is an island in the Caribbean with rural and urban areas identified in 78 municipalities, including the main island and several smaller islands, such as Mona, Vieques, and Culebra. According to the 2021 Decennial Census, the total population in Puerto Rico is 3,263,584, with a rural population of 136,992 [11]. Rural areas have histories, economies, and cultures that differ from those of urban areas and from one rural area to another [12]. Rural areas are defined according to the distance to the main urban centers. Rural areas can be known as isolated communities that present difficulties in access to basic services such as electricity, drinking water, communication, transportation, and health services.

The goal of the Hispanic Alliance for Clinical and Translational Research (Alliance) is to integrate clinical and translational research infrastructure and resources across a predominantly Hispanic U.S. jurisdiction to bridge the gaps between basic and clinical translational research to address severe health conditions that disproportionately affect the medically underserved and are highly prevalent among Hispanic populations. The Alliance is a partnership between three health Sciences Centers on the island of Puerto Rico: University of Puerto Rico Medical Sciences Campus, Universidad Central del Caribe, and Ponce Research Institute—Ponce Medical School Foundation, Inc.

Within the Alliance, the Community Engagement and Outreach Core (CoE) has undertaken the goal of catalyzing and supporting three meaningful conceptual frameworks, the Community Engagement Continuum (CEC) [13], the Community Academic Partnership Model (CAPs) [14], and the model of Communication for Social Change (CFSC) [15] to improve the health of the population in Puerto Rico. By following the main principles of these frameworks, we guarantee community involvement, impact, trust, and communication [16]. We are committed to improving health outcomes that affect Latino and Hispanic communities by nourishing engagement in research, starting with collaborations and ending in community-driven research efforts by integrating our academic partnership that fosters a meaningful community–academic partnership [17,18].

Understanding context is vital to devising approaches that yield community engagement. To improve the population’s health, we must know what are the health needs in Puerto Rico, particularly for those groups or communities experimenting more vulnerable situations. Our main research question was how to enhance and maintain community participation when identifying research priorities for health conditions prevalent in medically underserved populations. Therefore, the objectives of this research were to: (1) identify and prioritize key health-related issues needing attention in the isolated communities in Puerto Rico; and (2) develop strategies in terms of Policies, Programs, and Practices to address the identified community priorities. The article reports information related to targeted health priorities to the public and health policymakers. Additionally, the dissemination of the findings will stimulate the adoption of best practices for the development of an effective community–academic partnership.

## 2. Methods

CoE used the triangulation method for qualitative data by integrating the Colorado State University’s (CSU) Tri-ethnic Center Model, and the Delphi Technique for a better understanding of community health needs and priorities.

### 2.1. Data Collection

In March 2021, we invited members of isolated communities in Puerto Rico, health practitioners, and service providers to participate in a virtual community forum to dialogue with the CoE about current health priorities. Invitations and/or promotions were sent via social media, newspapers, and e-mails to local organizations, and messages to collaborators using the snowball approach. Promotions were also available on the websites of the Alliance and partnership universities. The study considered convenience sampling with the following inclusion criteria: residents, community leaders, community health practitioners, or service providers of rural areas in Puerto Rico, including the municipal islands of Culebra and Vieques. The exclusion criteria for this study were individuals with cognitive impairment who were unable to hold a conversation in the community forum and individuals younger than 21 years of age.

On 29 April 2021, the virtual forum was held via Zoom from 6:00 pm to 8:00 pm and conducted in Spanish since it is the predominant and primary language for the majority of Puerto Ricans. This qualitative research involved the collection of data through conversation, note-taking, recording, and the use of field notes that capture the researcher’s observations of the community health issues. The community forum format included: the state of the situation, antidote or thought-provoking questions, and results.

During the state of the situation, the epidemiological profile of the topic of discussion was presented using the available data. Participants were then distributed into three breakout rooms (6 participants, 6 participants, and 7 participants) to optimize dialogue between facilitators and participants. Each breakout room was facilitated by two members of the CoE. Facilitators in the three breakout rooms used the same thought-provoking questions (antidote) (Table 1) to guide participants through the process of evaluating a problem and prompting them to think about solutions. The thought-provoking questions were developed using the CSU Tri-ethnic Center Model to assess community readiness to address an issue on five key dimensions: community knowledge of the issue, community knowledge of efforts, community climate, leadership, and resources [19]. CoE members were acquainted with the questions to facilitate the discussion by encouraging dialogue and recording. Results or summaries were reported by each breakout room back to all forum participants. A list of health priorities was then developed by the participants.

### 2.2. Analysis

We used multiple triangulation techniques including data triangulation and investigator triangulation to drive higher confidence to describe health needs in isolated communities in Puerto Rico. The data were gathered from different perspectives due to the forum participants being people who hold different points of view or possess varying amounts of experience with the topic of isolated communities. Additionally, the virtual focus allowed the participation of individuals from different geographic regions of the island. Data collection was performed by six researchers and the analysis (data reduction and data displays) and conclusions (identification and prioritization of health-related issues needing attention in the isolated communities in Puerto Rico) involved nine researchers. In summary, qualitative notes were transcribed by the CoE team in Spanish and imported into a simple matrix using word processing software. The analysis was conducted by a multidisciplinary team with CoE members, who met weekly as the forums were conducted to clean notes and perform data quality control and analysis, thus ensuring a systematic, circular, and inductive approach between collection and analysis, which allowed the inductive emergence of strategies from the data collected in forums. The analysis included: a discussion of the integration of priorities into summarization processes to debrief forums transcription (capture information relevant to the populations and vulnerabilities discussed), the addition of listed priorities to the matrix, discussion of a plan to integrate priorities from other forums, and compilation of executive summaries per forum. CoE members met with partners to reflect on data and further the priority and alignment strategies, and data were reviewed by the CoE team and additional information was added as needed to clarify priorities.

### 2.3. Alliance Leaders and Advisory Boards’ Prioritization and List of Strategies

The Delphi Technique was used to prioritize the Alliance Health Focus Areas and to list the strategies to address health priorities. The Delphi Technique aims to develop expert-based judgment about an issue [20] and is beneficial when the problem at hand can benefit from collective, subjective judgments or decisions and when group dynamics do not allow for effective communication; for example, time differences, distance, and personality conflicts [21]. Alliance Leaders and Advisory Boards Prioritization consisted of an exercise to organize the community health priorities by importance and viability. Importance and viability were classified as: more important, less viable; more important, more viable; less important, less viable; and less important, more viable. The order of importance and viability was established using four dimensions: need for services, capacity, political commitment, and evidence-based practices. As a second exercise and part of a data validation process, the Alliance Leaders and the Advisory Boards developed strategies in terms of Policies, Programs, and Practices regarding the Alliance Leaders’ and Advisory Boards Prioritization. In summary, consensus was reached on priorities and emerging strategies in group meetings with the entire research team and partners.

## 3. Results

### 3.1. Community Health Priorities

Nineteen people participated in the forum, including participants from 13 municipalities (Figure 1) of Puerto Rico representing various sectors: citizens, community leaders, community health practitioners, and service providers (including advocates, academics, and researchers). The five community health-related priorities identified in the forum were: (1) access to health services (physical and mental); (2) older adults; (3) access to basic services; (4) preparedness for future natural disasters and emergencies; and (5) COVID-19 and access to vaccination (Table 2).

### 3.2. Alliance Leaders and Advisory Boards’ Prioritization and List of Strategies

The Alliance Leaders and the Advisory Boards categorized as “more important, more viable” the priorities of (1) preparedness for future natural disasters and (2) COVID-19 and access to vaccination. Therefore, they developed strategies to address these priorities (Table 3). The strategies developed to address the priorities were grouped into five areas of attention (Table 4).

## 4. Discussion

Our community forum generated unique insights into isolated communities’ needed health-related priorities in Puerto Rico by engaging citizens, community leaders, community health practitioners, and service providers (including advocates, academics, and researchers).

### 4.1. Isolated Communities’ Health-Related Priorities

The five priorities identified in the community forum were: (1) access to health services; (2) older adults; (3) access to basic services; (4) preparedness for future natural disasters/emergencies; and (5) COVID-19 and access to vaccination. The results show the barriers or needs that isolated communities must overcome to improve social determinants of health and reduce health disparities.

Unfortunately, the 2006–2016 economic crisis and several natural disasters have negatively impacted the health of Puerto Ricans. Additionally, the COVID-19 pandemic and lockdown measures in the U.S. have brought about nationwide physical, financial, and psychological consequences [22,23], including in Puerto Rico.

Over the last 10 years, Puerto Rico’s unemployment rate, together with the typically lowest pay scales in the USA, has been high, causing over 40% of the total population in Puerto Rico to live below the poverty level [24]. For low-income households, this translates into dependency on the Nutrition Assistance Program (NAP) (or Puerto Rico’s household food assistance program) and public health services for their medical care [25]. For fiscal year 2022, 1,556,788 individuals and 877,297 households in Puerto Rico participated in the NAP [26]. As of August 2021, Puerto Rico had enrolled 1.5 million individuals in Medicaid and the Children’s Health Insurance Program (CHIP) combined [25]. Compared to the 50 states and the District of Columbia, Puerto Rico receives less federal funding for health, even though it pays its full share of Social Security Insurance and Medicare taxes [27].

Although a generally poor economy has affected **access to health services**, after Hurricane Maria, the number of healthcare providers decreased by 6.5%, family physicians by 17.5%, and specialists by 8% [28]. Many medical professionals left Puerto Rico due to the financial crisis [28] and the decreasing reimbursement rates for physician services [29]. The Medicare reimbursement rate in Puerto Rico is 70% less than the reimbursement rate in the 50 U.S. states and the District of Columbia [29]. Additionally, structural changes in mental health and general healthcare services delivery from the public to a mostly privatized system showed limited quality, access, and utilization of mental health services [30].

Puerto Rico has 85 Community Health Centers (CHC) located throughout the island including Vieques and Culebra, particularly in high-need areas identified by the federal government for their high poverty, higher-than-average infant mortality, and shortage of physicians [31]. However, few municipalities in Puerto Rico have specialized hospitals [32]. For example, the only Diagnostic and Treatment Center that Vieques had was destroyed by Hurricane Maria in September 2017 and the construction process is yet to begin in 2023.

Over the last ten to twenty years, Puerto Rico has faced a decline in the younger population due to migration but has an increasing **number of older adults** (65 years and older) [24] who also suffer low socioeconomic circumstances and higher levels of disability, morbidity, and mortality from chronic disease compared to the rest of the population [33]. As noted by the Puerto Rico Department of Health, in 2010, chronic disease rates were, in general, significantly higher in households with annual incomes of $15,000 or less compared to more affluent households with annual incomes of $50,000 or more [33]. Heart disease, stroke, cancer diabetes, asthma, arthritis, and Alzheimer’s disease are the leading causes of disability and death in Puerto Rico [33]. About 18% of adults have serious difficulty walking or climbing stairs; 17% of adults have serious cognition limitations; 12% of adults have serious difficulty doing errands alone; 9% of adults have deafness or serious difficulty hearing; 19% of adults have serious vision limitations; and 6% of adults have self-care limitations [34]. These physical and cognitive disabilities represent constraints on the use of information and communication technologies (ICT) [35] for healthcare access. Other constraints on the use of technological devices for older individuals include the small size of the device, interface complexity of some devices, and difficulty of use and cost [35].

Although the population is aging, there is a serious shortage of long-term care facilities and support, and there is no coverage for long-term care under Puerto Rico’s Medicaid [36]. According to the 2021 Puerto Rico Family Caregiving Survey, two-thirds (67%) of Puerto Rican voters aged 45+ have unpaid family caregiving experience [37]. In Puerto Rico, caregivers are finding it difficult to deal with their own self-care (such as getting enough rest (61%) or socializing (47%), balancing work and caregiving (56%), emotional stress (56%), and even health problems (53%) [37].

Access to **basic services** is necessary to improve people’s lives. In the mid-to-late twentieth century, governmental policies favoring industrialization and tourism services indirectly contributed to significant declines in rural agriculture in Puerto Rico [38]. Currently, 85% of food is imported, and most agricultural outputs are exported [39]. The U.S.-imposed taxation on imported foods also raised the prices of produce [39], which means that Puerto Ricans must rely on expensive foods to survive. In 2015, the Behavioral Risk Factor System of the Department of Health found that in Puerto Rico, 33.2% of the population aged 18 and older suffered from food insecurity [40]. A survey (n = 1356) reports food insecurity in Puerto Rico at 38% before, and 40% since, the COVID-19 pandemic [41].

Water is a critical health determinant for sustainable development and the eradication of poverty and hunger, and is indispensable for human development, health, and well-being [42]. In Puerto Rico, the public potable water supply system is provided by the Puerto Rico Aqueduct and Sewer Authority (PRASA) and Non-PRASA systems [43]. PRASA is a public corporation and instrumentality of the Puerto Rican Government created under Act No. 40 of 1 May 1945, as amended [44], that manages and operates 97% of the potable water distribution system [45]. Non-PRASA systems serve rural or suburban housing areas and are not served by PRASA because they are community-operated water systems [46]. The Environmental Protection Agency (EPA) and the Puerto Rico Department of Health are wary of Non-PRASA systems because they are frequently non-compliant with the standards of the Safe Drinking Water Act (SDWA) [47].

Unfortunately, after Hurricane Maria wreaked destruction on Puerto Rico, and roughly one-third of the needed home reconstruction or repair has not yet been completed [48]. A critical factor that has complicated the recovery process is the high rates of informal housing present in Puerto Rico [49]. Informal housing includes houses without formal ownership of the land, built without permits, or without complying with building codes [50]. Additionally, in the “Homelessness Count of 2022” carried out in Puerto Rico, a total of 1016 individuals experiencing homelessness were counted; among them, 30.6% were between ages 50–65, and 9.1% were aged 65 or older [51].

**Natural and human disasters (or emergencies)** impact the population’s health, resulting in physical trauma, acute disease, and emotional trauma [52]. They also impact healthcare, which may increase the morbidity and mortality associated with chronic and infectious diseases [52]. Hurricanes Irma and Maria, in September 2017, caused severe damage to housing, infrastructure, and roads in Puerto Rico, and the island’s power grid failed for many months, causing long-lasting interruptions to essential services [53]. After Hurricane Maria, many communities in Puerto Rico were left without public water service and obtained their drinking water directly from local lakes, reservoirs, rivers, or tributaries without any filtration or purification treatment [54]. Later, in September 2022, Hurricane Fiona caused flooding in parts of the island and once again, Puerto Ricans suffered infrastructure damages. On 7 January 2020, Puerto Rico was impacted by yet another natural disaster, a series of earthquakes that caused structural damage throughout the southwest of the island [55,56].

The **COVID-19 pandemic** negatively impacted patients’ health while imposing substantial restrictions on access to healthcare, especially restricting access to services for disabled people [57]. Any reduction in access to often-minimal support networks or social engagement may have significant mental health consequences [57] and may prevent access to resources that enhance self-efficiency in disease management and resilience to disability [58].

### 4.2. Alliance Leaders and Advisory Boards’ Prioritization and List of Strategies

All health-related priorities identified in the forum of isolated communities in Puerto Rico are important, but Alliance Leaders and Advisory Boards understand that the CoE, together with Alliance resources, will work to improve preparedness for future natural disasters and emergencies, as well as COVID-19 and access to vaccination. For the priority of preparedness for future natural disasters and emergencies, ten strategies were developed (two at the policy level, six at the program level, and two at the practice level). Five strategies were developed to address COVID-19 and access to vaccination, including none at the policy level, two at the program level, and three at the practice level.

Strategies developed to address the priorities were grouped into five areas of attention. Areas that require more attention to minimize health-related disparities in Puerto Rico include (1) governmental support and welfare systems, including community-based organizations, (2) research in community settings, (3) education and training, (4) information dissemination, and (5) social contribution. The next steps are to: (1) develop the capacity of Alliance investigators, core staff, and community partners to address targeted health conditions through evidence-based research, community engagement, and mentoring and (2) improve the dissemination of research designs and findings to stimulate the adoption of best practices for community engagement.

## 5. Conclusions

Social determinants of health can produce health-related disparities and inequities between isolated communities in Puerto Rico that experience an intersectionality of factors or social determinants (e.g., socioeconomic class, disabilities, age, environment, and education access) that affect a wide range of health-related risks and outcomes. Recognizing barriers experienced by specific groups helps to develop interventions or strategies that are not focused solely on the healthcare sector.

As a goal, we expect that the research activities lined up in this publication will ensure community participation in decision-making to continue the identification of health-related priorities by engaging stakeholders from a diverse community sector. We understand that this will be made possible through the maturity of bi-directional and long-lasting community–academic collaborations and the integration of the community partners in research endeavors with Alliance cores and their collaborating academic institutions to build trusted relationships, decision-making, and shared responsibility for outcomes.

We will focus on sharing the findings across all sectors, raising awareness about needs, shining light on the sectors or populations experiencing social marginalization, disadvantage, limited access, or under-resourcing, and aligning the priorities identified with local and external resources at the individual, community, and policy levels. Promptly aligning research agendas will help address health-related community needs at a faster pace while maximizing resources, which we believe is imperative and necessary.

## Figures and Tables

**Figure 1 ijerph-20-06335-f001:**
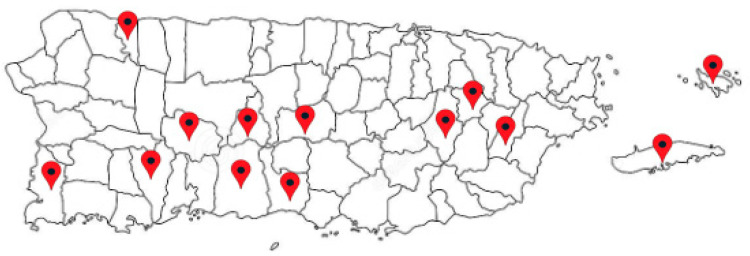
Municipalities of Puerto Rico represented in the forum.

**Table 1 ijerph-20-06335-t001:** Thought-provoking questions.

What is your experience as a member of an isolated community?
What is your experience working with an isolated population?
What are the top three (3) health priorities in your community?
Which social and environmental issues are present in your community?
Did these priorities (or needs) or issues arise with hurricanes Irma and Maria, earthquakes, or COVID-19? Have these priorities (or needs) or issues been in your community for years?
Have you felt listened to by the government? Does the government provide help to resolve some problems but not offer follow-up to their execution or implementation?
Identify some community strengths (or community capacity) to solve these priorities (or needs) or issues.
How do you bridge the gap between urban and rural areas?
What are the gaps in obtaining services and/or accessing services? What are the gaps in the delivery of services?

**Table 2 ijerph-20-06335-t002:** Priorities resulting from the community forum.

Community Health Priorities
**Access to health services** Lack of mental and physical healthcare access. Lack of knowledge about available health services and resources. **Older adults** Barriers to using technologies for healthcare access. Multiple health conditions (bedridden elderly, depression, renal disease, heart disease, cancer, Alzheimer’s). Older adults living alone. Older adults caring for their spouses or other family members. Senior care facilities are in deplorable conditions (without elevators, social workers, nurses, and others) or others were closed due to the COVID-19 pandemic. **Access to basic services** Food safety and nutrition due to low income. Lack of transportation by land or sea (ex. Vieques and Culebra) and road access. Water management in non-Puerto Rico Aqueduct and Sewer Authority ^1^ (non-PRASA) systems. Housing infrastructure. **Preparedness for future natural disasters and emergencies**
Education for prevention. Disaster-related stress. **COVID-19 and access to vaccination**
Lack of masks, education about the coronavirus disease, and access to COVID-19 vaccines.

^1^ Puerto Rico Aqueduct and Sewer Authority (PRASA) is a public corporation and instrumentality of the Puerto Rican Government that owns and operates the public water supply and wastewater systems.

**Table 3 ijerph-20-06335-t003:** Strategies identified to address the priority of preparedness for future natural disasters and emergencies and COVID-19 and access to vaccination.

Priorities	Strategies Identified to Address the Priorities
**Preparedness for future natural disasters and emergencies**	**Policies**
	Evaluate the integration of emerging key organizations and the central government to prepare for future disasters or emergencies. Review the centralized plan for emergency/disaster preparedness and management.
	**Programs**
	Develop an educational campaign to help prepare for natural disasters in collaboration with the Puerto Rico Department of Health, American Red Cross, Puerto Rico Science, Technology, and Research Trust, and the Federal Emergency Management Agency. Develop guidance on special considerations for people (children and adults) with functional diversity and their families in natural disasters/emergencies planning and response. Create a repository of resources and services for disaster/emergency preparedness, management, response, and recovery in Puerto Rico. Promote “Disaster Medicine” as an area of interest to receive proposals. Help in the dissemination of available information and training programs on disaster preparedness. Create and develop citizen participation programs.
	**Practices**
	Conduct studies to explore the level of preparedness for natural disasters in isolated communities. Carry out “Community Outreach and Assessment” in communities at greatest risk of natural disasters or emergencies in Puerto Rico.
**COVID-19 and access to vaccination**	**Policies**
	No Public Policy strategy was developed for this priority.
	**Programs**
	Disseminate health promotion strategies to communities on topics including COVID-19 vaccination, prevention, and control in collaboration with the Puerto Rico Department of Health, Walgreens, Centers for Disease Control and Prevention, and others. Promote the creation of a repository including resources and organizations involved in the response to COVID-19 and serve as a liaison to improve access to communities in need. Identify key people and community leaders and/or service providers.
	**Practices**
	Promote studies related to COVID-19 and vaccination in specific communities. Develop a registry of COVID-19 studies in communities in Puerto Rico, at least in Alliance institutions. Promote coordination with vaccination centers of the institutions affiliated with the Alliance to reach isolated communities.

**Table 4 ijerph-20-06335-t004:** Alignment of strategies within the areas of governmental support and welfare systems; research in the community settings, education and training, information dissemination, and social contributions.

Strategies Identifies to Address the Priorities	Areas				
	1 Governmental/ Welfare Systems	2 Research in Community Settings	3 Education and Training	4 Information Dissemination	5 Social Contributions
**Preparedness for future natural disasters and emergencies**					
**Polices**					
Evaluate the integration of emerging key organizations and the central government to prepare for future disasters or emergencies.	**X**				**X**
Review the centralized plan for emergency/disaster preparedness and management.	**X**	**X**			
**Programs**					
Develop an educational campaign to help prepare for natural disasters in collaboration with the Puerto Rico Department of Health, American Red Cross, Puerto Rico Science, Technology and Research Trust, and the Federal Emergency Management Agency.	**X**		**X**		
Develop guidance on special considerations for people (children and adults) with functional diversity and their families in natural disasters/emergencies planning and response.	**X**		**X**		
Create a repository of resources and services for disaster/emergency preparedness, management, response, and recovery in Puerto Rico.	**X**			**X**	**X**
Promote “Disaster Medicine” as an area of interest to receive proposals.		**X**			
Help in the dissemination of available information and training programs on disaster preparedness.	**X**		**X**	**X**	
Create and develop citizen participation programs.	**X**		**X**		**X**
**Practices**					
Conduct studies to explore the level of preparedness for natural disasters in isolated communities.		**X**			**X**
Carry out “Community Outreach and Assessment” in communities at greatest risk of natural disasters or emergencies in Puerto Rico.		**X**			
**COVID-19 and access to vaccination**					
**Programs**					
Dissemination of health promotion strategies to communities on topics including COVID-19, vaccination, prevention, and control in collaboration with the Puerto Rico Department of Health, Walgreens, Centers for Disease Control and Prevention, and others.	**X**			**X**	
Promote the creation of a repository including resources and organizations involved in the response to COVID-19 and serve as a liaison to improve access to communities in need. Identify key people and community leaders and/or service providers.	**X**			**X**	**X**
**Practices**					
Promote studies related to COVID-19 and vaccination in specific communities.		**X**			
Develop a registry of COVID-19 studies in communities in Puerto Rico, at least in Alliance institutions.		**X**		**X**	
Promote coordination with vaccination centers of the institutions affiliated with the Alliance to reach isolated communities.					**X**

## Data Availability

No new data were created.

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
