# Peer review of "Community Forums as Amplifiers of Communities’ Voices: Isolated Communities in Puerto Rico"

_ijerph, 2023, doi:10.3390/ijerph20146335_

Round 1
Reviewer 1 Report
Thank you for the opportunity to review the paper “Community Forums as Amplifiers of Communities’ Voices: Isolated Communities in Puerto Rico”. This paper shed light on the needs of marginalized communities in Puerto Rico and identifies strategies to better meet the needs of these communities. This paper addresses a very important topic and can benefit from further connecting the introduction/literature review with the results/discussion and clarifications throughout all sections.
Introduction
· Under Introduction on p.5: the authors presented the implications of social isolation and loneliness and then shifted to the isolated communities in Puerto Rico. This transition is a bit abrupt. Loneliness/isolation experienced by individuals is different from isolation/segregation at a community level. How are isolated communities defined and identified? Are the authors assuming that all individuals in isolated communities are isolated and lonely? The authors might benefit from expanding their discussion on the lack of social determinants of health systematically in certain communities. I am not sure if the texts on individuals’ isolation/loneliness are relevant to this paper because the forum intends to shed light on community needs/resources instead of individual loneliness/isolation.
· Relatedly, please further explain how communities and individuals were selected. What were the inclusion and exclusion criteria?
· P. 5 “On April 29, 2021, a community forum was held on the topic of Isolation, focusing on 90 residents of rural areas - including the municipal islands of Culebra and Vieques - elderly people living alone or caring for other elderly and/or sick people and people with disabilities or chronic diseases.” This section might fit better under the Methods section. Also, What is the sampling strategy?
· P.5 “Loneliness is a unique condition in which an individual perceives himself or herself to be socially isolated even when among other people 58 and has been associated with social isolation, depression, introversion, or poor social skills”. It is confusing that the authors defined loneliness as subjective social isolation. Loneliness is often defined as a subjective emotional state resulting from dissatisfaction with the quality and quantity of social relationships. There is chronic and transitory loneliness. What is the author(s) interested in addressing here?
Method
· Please consider adding an analysis section. The information on the triangulation method can be better organized under the analysis section. What are the methods to ensure the rigor and credibility of the qualitative themes?
· “The community forum format included: state of the situation, antidote or thought-provoking questions, and results.” Is it possible to add some example questions asked in the forum? What do thought-provoking questions mean?
· What language was the forum conducted? How does that affect research participation?
Results
· The community forum was held on the topic of Isolation, but I have a hard time understanding how the findings are related to the isolation of the community or individuals.
· The authors used the wording “elderly population” several times. Please review the language recommendations from reframe aging initiative and consider using the term “older adults”.
Discussion
· Please explain how strategies identified to address the community priorities can address isolation at the community or individual level.
Thank you!
Moderate English Language editing recommended.
Author Response
We appreciate the time and effort that you have dedicated to providing your valuable feedback on our manuscript. Here is a point-by-point response to your comments and concerns.
- Under Introduction on p.5: the authors presented the implications of social isolation and loneliness and then shifted to the isolated communities in Puerto Rico. This transition is a bit abrupt. Loneliness/isolation experienced by individuals is different from isolation/segregation at a community level. How are isolated communities defined and identified? Are the authors assuming that all individuals in isolated communities are isolated and lonely? The authors might benefit from expanding their discussion on the lack of social determinants of health systematically in certain communities. I am not sure if the texts on individuals’ isolation/loneliness are relevant to this paper because the forum intends to shed light on community needs/resources instead of individual loneliness/isolation. Response: We deleted the information in the Introduction section related to individuals’ isolation/loneliness.
-
- We added this information in the introduction section (lines 57-71) “Disparities in income, education, occupational status, or wealth are translated into a measure of social hierarchy called socioeconomic status (SES) [6-7]. SES has been viewed as the driver of discrimination and racial inequities in health [7]. Healthcare discrimination occurs at both individual and structural level, due to race, gender, and other elements [7-8]. Discrimination at the individual level includes negative interactions between a patient and a healthcare provider and structural discrimination is seen in the form of residential segregation for geographic and economic reasons [7-8]. According to an intersectionality perspective, health inequities are the results of different social locations (e.g., race/ethnicity, Indigeneity, gender, class, sexuality, geography, age, disability/ability, migration status, religion), power relations (e.g., laws, policies, state governments and other political and economic unions, religious institutions, media) and experiences [9]. Also, the historical trauma presents that populations historically subjected to long-term, collective trauma (or traumatic experience) transmit these traumas to subsequent generations, causing them a higher prevalence of adverse effects, including an impact on physical and mental health [10]. Also we added this sentence (lines 78-81) “Rural areas can be known as isolated communities that present difficulties in the access to basic services such as electricity, drinking water, communication, transportation, and health services”.
-
2. Relatedly, please further explain how communities and individuals were selected. What were the inclusion and exclusion criteria?
Response: We added this information under the Methods section “The study considered convenience sampling with the following inclusion criteria: residents, community leaders, community health practitioners or services providers of rural areas of Puerto Rico – including the municipal islands of Culebra and Vieques. The exclusion criteria for this study were individuals with cognitive impairment who were unable to hold a conversation in the community forum and individuals younger than 21 years of age”.
3. P. 5 “On April 29, 2021, a community forum was held on the topic of Isolation, focusing on 90 residents of rural areas - including the municipal islands of Culebra and Vieques - elderly people living alone or caring for other elderly and/or sick people and people with disabilities or chronic diseases.” This section might fit better under the Methods section. Also, what is the sampling strategy? Response: We deleted this information “On April 29, 2021, a community forum was held on the topic of Isolation, focusing on 90 residents of rural areas - including the municipal islands of Culebra and Vieques - elderly people living alone or caring for other elderly and/or sick people and people with disabilities or chronic diseases.”
We added this information under the Methods section “In March 2021, we invited members of isolated communities in Puerto Rico, health practitioners, and service providers to participate in a virtual community forum to dialogue with the CoE about current health priorities. Invitations and/or promotions were sent via social media, newspapers, and e-mails to local organizations, and messages to collaborators using the snowball approach. Promotions were also available on the websites of the Alliance and partnership universities. The study considered convenience sampling with the following inclusion criteria: residents, community leaders, community health practitioners, or service providers of rural areas in Puerto Rico – including the municipal islands of Culebra and Vieques. The exclusion criteria for this study were individuals with cognitive impairment who were unable to hold a conversation in the community forum and individuals younger than 21 years of age. On April 29, 2021, the virtual forum was held via Zoom from 6:00 pm to 8:00 pm, and conducted in Spanish since it is the predominant and primary language for the majority of Puerto Ricans”.
4. P.5 “Loneliness is a unique condition in which an individual perceives himself or herself to be socially isolated even when among other people 58 and has been associated with social isolation, depression, introversion, or poor social skills”. It is confusing that the authors defined loneliness as subjective social isolation. Loneliness is often defined as a subjective emotional state resulting from dissatisfaction with the quality and quantity of social relationships. There is chronic and transitory loneliness. What is the author(s) interested in addressing here? Response: We deleted this information.
5. Method
-
- Please consider adding an analysis section. The information on the triangulation method can be better organized under the analysis section. What are the methods to ensure the rigor and credibility of the qualitative themes?
- Response: We added an Analysis section with this information (lines 147-169) “We used multiple triangulation techniques including data triangulation and investigator triangulation to drive higher confidence to describe health needs in isolated communities in Puerto Rico. The data was gathered from different perspectives due to the forum participants being people who hold different points of view or possess varying amounts of experience with the topic of isolated communities. Also, the virtual focus allowed the participation of individuals from different geographic regions of the island. Data collection was done by six researchers and the analysis (data reduction and data displays) and conclusions (identification and prioritization of health-related issues needing attention in the isolated communities in Puerto Rico) involved nine researchers. In summary, qualitative notes were transcribed by the CoE team in Spanish and imported into a simple matrix using a word processing software. The analysis was conducted by a multidisciplinary team with CoE members, who met weekly as the forums were conducted, to clean notes, perform data quality control and analysis, thus ensuring a systematic, circular, and inductive approach between collection and analysis which allowed the inductive emergence of strategies from the data collected in forums. The analysis included: a discussion of the integration of priorities into summarization processes debrief forums transcription (capture information relevant populations and vulnerabilities discussed), the addition of listed priorities to the matrix, discussion of plan to integrate priorities from other forums, and compilation of executive summaries per forum. CoE members meet with partners to reflect on data and further the priority and alignment strategies, data was reviewed by the CoE team and additional information was added as needed to clarify priorities”.
- Please consider adding an analysis section. The information on the triangulation method can be better organized under the analysis section. What are the methods to ensure the rigor and credibility of the qualitative themes?
6. The community forum format included: state of the situation, antidote or thought-provoking questions, and results.” Is it possible to add some example questions asked in the forum? What do thought-provoking questions mean?Response: We added Table 1 on line 170. Thought-provoking questions are questions for assessing, diagnosing, and acting on a problem or topic.
Table 1. Thought-provoking questions.
|
1 |
What is your experience as a member of an isolated community? |
|
2 |
What is your experience working with an isolated population? |
|
3 |
What are the top three (3) health priorities in your community? |
|
4 |
Which social and environmental issues are present in your community? |
|
5 |
Did these priorities (or needs) or issues arise with hurricanes Irma and Maria, earthquakes, or COVID-19? Have these priorities (or needs) or issues been in your community for years? |
|
|
Have you felt listened by the government? Does the governments provided help to resolve some problems but do not offer follow-up to their executions or implementations? |
|
6 |
Identify some community strengths (or community capacity) to solve these priorities (or needs) or issues. |
|
7 |
How do you bridge the gap between urban and rural areas? |
|
8 |
What are the gaps in obtaining services and/or accessing services? What are the gaps in the delivery of services? |
7. What language was the forum conducted? How does that affect research participation? Response: We added this information in lines 128-130 “On April 29, 2021, the virtual forum was held via Zoom from 6:00 pm to 8:00 pm, and conducted in Spanish since it is the predominant and primary language for the majority of Puerto Ricans.”.
8. Results
-
- The community forum was held on the topic of Isolation, but I have a hard time understanding how the findings are related to the isolation of the community or individuals. Response: Results present barriers or needs that isolated communities (or rural communities) must overcome to improve social determinants of health and reduce health disparities. Also, the study presents strategies for reducing health disparities in isolated communities.
- The authors used the wording “elderly population” several times. Please review the language recommendations from reframe aging initiative and consider using the term “older adults”. Response: We changed elderly population to older adults.
- Discussion
- Please explain how strategies identified to address the community priorities can address isolation at the community or individual level. Response: We added this sentence in discussion (lines 243-245) “The results show the barriers or needs that isolated communities must overcome to improve social determinants of health and reduce health disparities”.

Reviewer 2 Report
Formatting issues, authors need to review and ensure that all of the formatting is consistent throughout the manuscript.
|
Manuscript Number: |
IJERPH-2385350 |
|
Overall manuscript |
Formatting issues, authors need to review and ensure that all of the formatting is consistent throughout the manuscript.
There are several minor grammatical errors that need to be addressed. |
|
Abstract: |
|
|
Line 29 |
Remove the semicolon and add “and” |
|
Lines 35-39 |
The sentences that begin with “The Alliance Leaders…” and “The strategies…” are awkward sentences and need to be reworded. |
|
Introduction: |
|
|
References 1 & 2 |
Both need to be updated to refer to the HP2030 website instead of the CDC website. |
|
Lines 45-47 |
Need to paraphrase the SDoH definition better, it is almost word for word the same as listed on the HP2030 website. The rearranging of word order made the sentence awkward and the meaning hard to decipher. |
|
Line 47 |
HP 2030 – not 2020 |
|
Line 51 |
No need to place the HHS acronym if this is only place in the manuscript that you mention the Department. |
|
Lines 69-70 |
Either make the word Hispanic plural or the word Latinos singular
Do the authors mean the largest racial/ethic minority group? |
|
Results: |
|
|
Table 2 |
Formatting issues – there is a bold line below Preparedness Practices
The title of the table needs to be at the same left margin as the table.
The lines for the Strategies going all the way to the left margin is distracting and not visually pleasing. Can the lines stop at the left edge of the right column and not go across both columns?
Need to specific that it is the American Red Cross. There are numerous Red Cross and Red Crescent national societies globally.
Notes need to be provided for the table to explain acronyms such as CDC |
|
Discussion |
|
|
Line 186 |
NAP was already defined in line 183, the author did not need to redefine it.
|
|
Lines 217 & 218 |
Spacing between words |
|
Table 2 |
Should be labeled Table 3
Title of table should go to the left margin of the table
Formatting issues similar to Table 2 in the Results
Notes need to be provided for the table to explain acronyms such as CDC |
|
Limitations |
Where are these located?? |
|
Conclusion |
Needs to be expanded. |
There are several minor grammatical errors that need to be addressed.
Author Response
We appreciate the time and effort that you have dedicated to providing your valuable feedback on our manuscript. Here is a point-by-point response to the your comments and concerns.
- Overall manuscript – Formatting issues, authors need to review and ensure that all of the formatting is consistent throughout the manuscript. There are several minor grammatical errors that need to be addressed. Response: We reviewed the format throughout the manuscript and a professional translator reviewed the document.
Abstract
-
- Line 29 – Remove the semicolon and add and. Response: We removed the semicolon (; ) and add and.
- Lines 35-39 – The sentences that begin with “The Alliance Leaders…and “The strategies…” are awkward and need to be reworded. Response: We reworded lines 35-38 –“ The Alliance Leaders and Advisory Boards understand that we will work with the priorities of preparedness for future natural disasters/emergencies and COVID-19 and access to vaccination. Fifteen strategies were developed for these priorities and were grouped into five areas that require more attention in order to reduce health disparities.”.
Introduction
- References 1 & 2 – Both need to be updated to refer to the HP2030 website instead of the CDC website. Response – References 1 & 2 were updated:
- Healthy People 2030. Social determinants of health. Available online: https://health.gov/healthypeople/priority-areas/social-determinants-health (accessed on 12 June 2023).
- Healthy People 2030. Healthy People 2030 Framework. Available online: https://health.gov/healthypeople/about/healthy-people-2030-framework (accessed on 12 June 2023).
2. Lines 45-47 – Need to paraphrase the SDoH definition better, it is almost word for word the same as listed on the HP2030 website. The rearranging of word order made the sentence awkward and the meaning hard to decipher. Response: We changed the sentence - “Social determinants of health (SDOH) are the social conditions in which people born, grow, live, learn, work, play, and age that contributes to health disparities and inequities”.
Line 47 – HP 2030- not 2020
Response – We changed Healthy People 2020 for Healthy People 2030.
Line 51 – No need to place the HHS acronym if this is only placed in the manuscript that you mention the Department.
Response: We deleted the acronym HHS.
Lines 69-70 – Either make the word Hispanic plural or the word Latino singular. Do the authors mean the largest racial/ethnic minority group? Response: We changed Hispanic for Hispanics. Yes, Hispanics or Latinos are the largest racial/ethnic minority group in the US.
Results
Table 2 -Formatting issues – there is a bold line below Preparedness Practices Response: We revised the table, which now is labeled Table 3.
The title of the table needs to be at the same left margin as the table. Response: The title was placed at the same left margin as the table.
The lines for the Strategies going all the way to the left margin are distracting and not visually pleasing. Can the lines stop at the left edge of the right column and not go across both columns? Response: We stopped the lines at the left edge of the right column.
Need to specify that it is the American Red Cross. There are numerous Red Cross and Red Crescent national societies globally, Response: We specified American Red Cross.
Notes need to be provided for the table to explain acronyms such as CDC. Response: We added the explanation of the CDC acronym in the table.
Discussion
Line 186 – NAP was already defined in line 183, the author did not need to redefine it. Response: We deleted the definition of NAP in line 186 (now line 256).
Lines 217 & 218 Spacing between words. Response: We revised the spaces in the sentence: These physical and cognitive disabilities represent constraints on the use of information and communication technologies (ICT) [35] for healthcare access (now lines 287-289).
Table 2 – Should be labeled Table 3
Response: We revised Tables numbers.
Title of the table should go to the left margin of the table
Response: The title of Table 3 was placed at the same left margin as the table.
Formatting issues like Table 2 in the results.
Response: We formatted Table 2 (now Table 4).
Notes need to be provided for the table to explain acronyms such as CDC.
Response: We added the explanation of the CDC acronym in the table.
Limitations – Where are these located?
Response: The COVID-19 pandemic prompted us to look for new ways to engage people. The activities and meetings through virtual platforms made it possible to overcome geographical limits and a broader participation beyond what was initially proposed.
Conclusion – Needs to be expanded.
Response: We reviewed the conclusion “Social determinants of health can produce health-related disparities and inequities between isolated communities in Puerto Rico that experience intersectionality of factors or social determinants (e.g., socioeconomic class, disabilities, age, environment, and education access) that affect a wide range of health-related risks and outcomes. Recognizing barriers experienced by specific groups helps to develop interventions or strategies that are not focused solely on the healthcare sector.
As a goal, we expect that the research activities lined up in this publication will ensure community participation in decision-making to continue the identification of health-related priorities by engaging stakeholders from a diverse community sector. We understand that this will be made possible through the maturity of bi-directional and long-lasting community-academic collaborations and the integration of the community partners in research endeavors with Alliance cores and its collaborating academic institutions to builds trusted relationships, decision-making, and shared responsibility for outcomes.
We will focus on sharing the findings across all sectors, raising awareness about the needs, shining light on the sectors or populations experiencing social marginalization, disadvantage, limited access, or under-resourced, and aligning the priorities identified with local and external resources at the individual, community, and policy level. Promptly aligning research agendas will help address health-related community needs at a faster pace while maximizing resources, which we believe is imperative, and necessary”.

Reviewer 3 Report
1. I suggest expanding the background of this study.
2. It would be convenient to delve into some model or some models that explain the Development of the psychological dimensions of the problems of isolation of the communities and the loneliness of the people of those communities in Puerto Rico and in other parts of Latin America. Likewise, the approach or attention to these problems. It can be accompanied with some scheme.
3. The research questions must be clearly specified
4. The objective of this manuscript must be specified.
5. I suggest changing the format of a community intervention report to a scientific article on a community intervention, especially from line 97 to line 142 (pages 4 and 5), and also in the results.
6. In discussion and based on the results, they could propose some scheme that characterizes the intervention and the theoretical derivations that can be generated from this study.
7. In discussion, statistical data is added, it would be better if they were put in the Results section. And leave the discussion to analyze and compare these results.
8. Table 2 should be announced in the full text of the article. Perhaps it is better that I go before the Discussion itself. Table 3 is announced in the text but, the title says Table 2.
Author Response
We appreciate the time and effort that you have dedicated to providing your valuable feedback on our manuscript. Here is a point-by-point response to your comments and concerns.
Comments from Reviewer 3
- I suggest expanding the background of this study. Response: As a recommendation of reviewer 1, we deleted the information about individuals’ isolation/loneliness and added this information “Disparities in income, education, occupational status, or wealth are translated into a measure of social hierarchy called socioeconomic status (SES) [6-7]. SES has been viewed as the driver of discrimination and racial inequities in health [7]. Healthcare discrimination occurs at both individual and structural levels, due to race, gender, and others [7-8]. Discrimination at the individual level includes negative interactions between a patient and a healthcare provider and structural discrimination is seen in the form of residential segregation for geographic and economic reasons [7-8]. According to an intersectionality perspective, health inequities are the results of different social locations (e.g., race/ethnicity, Indigeneity, gender, class, sexuality, geography, age, disability/ability, migration status, religion), power relations (e.g., laws, policies, state governments and other political and economic unions, religions institution, media) and experiences [9]. Also, the historical trauma presents that populations historically subjected to long-term, collective trauma (or traumatic experience) transmit these traumas to subsequent generations, causing them a higher prevalence of adverse effects, including an impact on physical and mental health [10].
2. It would be convenient to delve into some model or models that explain the Development of the psychological dimensions of the problems of isolation of the communities and the loneliness of the people of those communities in Puerto Rico and other parts of Latin America. Likewise, the approach or attention to these problems. It can be accompanied by some scheme. Response: Lines 57-71 depict our theoretical approaches (science equity, intersectionality, and historical trauma). Nevertheless, our science is also guided by additional frameworks such as the Community Engagement Continuum, Community-Academic partnerships, and the Communication for Social Change which have been added to the introduction to expand as requested (lines 90-99).
3. The research questions must be specified. Response: We added the overall research question of the Community Outreach Core of the Alliance that allowed for this research activity to happen. See in lines 94-96 “Our main research question was how to enhance and maintain community participation in identifying research priorities for health conditions prevalent in medically underserved populations?”
4. The objective of this manuscript must be specified. Response: The article reports information related to targeted health priorities to the public and health policymakers. Also, the dissemination of the findings will stimulate the adoption of best practices for the development of an effective community-academic partnership (lines 108-111).
5. I suggest changing the format of a community intervention report to a scientific article on community intervention, especially from line 97 to line 142 (pages 4 and 5), and also in the results. Response: We changed the forms of presenting the information.
6. In discussion and based on the results, they could propose some scheme that characterizes the intervention and the theoretical derivations that can be generated from this study. Response. We added in lines 352-363 details on these endeavors. Neverlethess, the result of this study is a guideline to align research plans to community needs.
7. In discussion, statistical data is added, it would be better if they were put in the Results section. And leave the discussion to analyze and compare these results. Response: The statistical data added in the discussion is used to explain results, but it is not part of the research results.
8. Table 2 should be announced in the full text of the article. Perhaps I should go before the Discussion itself. Table 3 is announced in the text but, the title says Table 2. Response: We revised Tables numbers. Tables were announced in the full text of the article.

Round 2
Reviewer 1 Report
The authors have thoroughly addressed my comments. Thanks.
Reviewer 2 Report
Thank you for the updates to your manuscript.